# Breastfeeding outcomes in late preterm infants: A multi-centre prospective cohort study

Amy Keir[1,2,3,4]*, Alice Rumbold[1,2,4], Carmel T. Collins[1,2], Andrew J. McPhee[1], Jojy Varghese[5], Scott Morris[6], Thomas R. Sullivan[1,7], Shalem Leemaqz[1], Philippa Middleton[1], Maria Makrides[1,2], Karen P. Best[1,2]

**1** SAHMRI Women and Kids, South Australian Health and Medical Research Institute, Adelaide, South Australia, **2** Adelaide Medical School, The University of Adelaide, Adelaide, South Australia, **3** Department of Neonatal Medicine, Women's and Children's Hospital, North Adelaide, South Australia, **4** Robinson Research Institute, The University of Adelaide, Adelaide, South Australia, **5** Lyell McEwin Hospital, Elizabeth Vale, South Australia, **6** Department of Neonatology, Flinders Medical Centre, Bedford Park, South Australia, Australia, **7** School of Public Health, The University of Adelaide, Adelaide, South Australia

* amy.keir@adelaide.edu.au

## Abstract

### Objectives

To describe (1) infant feeding practices during initial hospitalisation and up to 6 months corrected age (CA) in infants born late preterm with mothers intending to breastfeed, (2) the impact of early feeding practices on hospital length of stay and (3) maternal and infant factors associated with duration of breastfeeding.

### Methods

We conducted a prospective cohort study of infants born at $34^{+0}$ to $36^{+6}$ weeks gestational age during 2018–2020. Families were followed up until the infant reached 6 months of age (corrected for prematurity). Feeding practices during the birth hospitalisation, length of initial hospital stay, and the prevalence of exclusive or any breastfeeding at 6 weeks, 3 months, and 6 months CA were examined. Associations between maternal and infant characteristics and breastfeeding at 6 weeks, 3 months and 6 months CA were assessed using multivariable logistic regression models.

### Results

270 infants were enrolled, of these, 30% were multiple births. Overall, 78% of infants received only breastmilk as their first feed, and 83% received formula during the hospitalisation. Seventy-four per cent of infants were exclusively breastfed at discharge, 41% at 6 weeks CA, 35% at 3 months CA, and 29% at 6 months CA. The corresponding combined exclusive and partial breastfeeding rates (any breastfeeding) were 72%, 64%, and 53% of babies at 6 weeks CA, 3 months CA, and 6 months CA, respectively. The mean duration of hospitalisation was 2.9 days longer (95% confidence interval (CI) 0.31, 5.43 days) in infants who received any formula compared with those receiving only breastmilk (adjusted for GA, maternal age, multiple birth, site, and neonatal intensive care unit admission). In

**Data Availability Statement:** Data cannot be shared publicly because of restrictions placed as a condition of ethical approval from the Women's and Children's Health Network Human Research

Ethics Committee. Our study contains human research participant data with potentially identifying patient information. When this study was designed it was not our standard practice to obtain participant consent for data sharing. We are therefore unable to publicly share this data as we have not sought participant consent. We will submit any requests for data to our ethics committee and make data sets available once approved. Data are available from the Women's and Children's Health Network Human Research Ethics Committee (contact via the WCHN Research Governance Officer at HealthWCHNResearch@sa.gov.au) for researchers who meet the criteria for access to confidential data.

**Funding:** AK, TS, CTC, PM and MM receive funding from the Australian National Health and Medical Research Council (NHMRC: www.nhmrc.gov.au): APP1161379, APP1173576, APP1132596, App1172870 and APP1154912, respectively. KPB is supported by an MS McLeod Post-doctoral Fellowship from the Women and Children's Hospital Foundation (WCHF: wchfoundation.org.au). The contents of this paper are solely the responsibility of the individual authors and do not reflect the views of the NHMRC or any other funding body. The funders had no role in study design, data collection and analysis, decision to publish, or manuscript preparation.

**Competing interests:** The authors have declared that no competing interests exist.

multivariable models, receipt of formula as the first milk feed was associated with a reduction in exclusive breastfeeding at 6 weeks CA (odds ratio = 0.22; 95% CI 0.09 to 0.53) and intention to breastfeed >6 months with an increase (odds ratio = 4.98; 95% CI 2.39 to 10.40). Intention to breastfeed >6 months remained an important predictor of exclusive breastfeeding at 3 and 6 months CA.

## Conclusions

Our study demonstrates that long-term exclusive breastfeeding rates were low in a cohort of women intending to provide breastmilk to their late preterm infants, with approximately half providing any breastmilk at 6 months CA. Formula as the first milk feed and intention to breastfeed >6 months were significant predictors of breastfeeding duration. Improving breastfeeding outcomes may require strategies to support early lactation and a better understanding of the ongoing support needs of this population.

## Introduction

Every year around 15 million infants are born preterm [1]. Late preterm infants born between $34^{+0}$ to $36^{+6}$ weeks gestation make up approximately 7% of the infant population and about 70–75% of all preterm births [2, 3]. Late preterm infants are often considered functionally full term because of their size and because they are generally clinically stable. They are, however, born physiologically and metabolically immature. Consequently, late preterm infants have higher morbidity rates, prolonged hospitalisation and re-hospitalisation, and healthcare costs than term infants [4].

Breastmilk is the optimal nutrition to support overall growth and development, gut maturation and immune protection. Complications relating to poor feeding are major contributors to the health burden in this population and one of the leading causes of extended hospital stay and readmission [5]. Yet, there is very little evidence to guide nutritional management of this group, as research in neonatal nutrition has primarily focussed on improving outcomes in very preterm infants born less than 32 weeks gestation [6]. Notably, contemporary evidence about breastfeeding outcomes for infants born late preterm is lacking. The limited available evidence suggests breastfeeding rates in the late preterm population are lower than in infants born at term [7, 8]. A key limitation of existing studies is the focus on breastfeeding status at discharge from the hospital rather than longer-term breastfeeding outcomes [9]. Further, in term infants, early exposure to formula may interfere with successful long-term breastfeeding [10] but this has not been fully explored in late preterm infants. Understanding the progression of breastfeeding in this population beyond discharge, the influence of early feeding practices on hospital stay, and the key factors associated with breastfeeding duration will help identify the areas where targeted support may help mothers sustain breastfeeding.

Our study aimed to describe feeding practices in infants born late preterm, the maternal and infant factors associated with longer duration of breastfeeding and the impact of early formula feeding on hospital length of stay.

## Methods

### Study design

Prospective cohort study of feeding patterns to 6 months corrected age of infants born at $34^{+0}$ to $36^{+6}$ weeks gestation (late-preterm).

## Setting

Research personnel identified late-preterm infants during their birth admission in the neonatal unit or postnatal ward of the Women's and Children's Hospital (WCH), Flinders Medical Centre (FMC) and Lyell McEwin Hospital (LMH). All centres are in metropolitan South Australia and represent the three major perinatal centres in the state. Women were recruited between 8th August 2018 and 18th September 2019 and data collection was finalised on 24th April 2020.

## Participants

Mothers of infants born $34^{+0}$ to $36^{+6}$ weeks gestation, intending to breastfeed and residing in South Australia, were eligible for inclusion up to 42 weeks post-menstrual age (PMA). Infants with a lethal congenital anomaly and/or not expected to survive to discharge home were excluded, as were women choosing to formula feeding their babies exclusively at the time of screening. Women provided written or electronic informed consent to participate.

## Data collection

At enrolment, data were collected on infants (gestational age, plurality, sex, weight, length and head circumference at birth, short-term clinical outcomes), mothers (age, born in Australia or overseas, whether they identify as Aboriginal or Torres Strait Islander, postcode, type of delivery), and maternal feeding intent before to delivery and at enrolment (breastfeeding, formula, mixed feeding, or undecided). Data were collected from infant medical records on date and type of first enteral feed (maternal breastmilk, donor breast milk or formula), type of enteral feeds at discharge, type and method of feeding on discharge home, and mother and infant hospital length of stay. Maternal surveys were designed using a Research Electronic Data Capture (REDCap) database (https://www.project-redcap.org/) [11, 12] and sent to participants by text message to their mobile phone or by email. Surveys were sent weekly until infants reached 42 weeks corrected age (CA) and then at 6 weeks CA, 3 months CA, and 6 months CA.

Data were collected about breastfeeding practices at each time point and included current breastfeeding status and feeding method. All data were collected and managed using the REDCap database hosted on secure servers by the South Australian Health and Medical Research Institute.

## Outcomes

Key outcomes were exclusive breastfeeding at 6 weeks, 3 months, and 6 months of age, corrected for prematurity, and any breastfeeding (exclusive or mixed feeding) at these time points. Any breastfeeding was defined as a baby receiving any breast milk at each time point and in the preceding week. Exclusive breastfeeding was defined as a baby receiving only breastmilk (with the exception of medications, oral rehydration solutions or vitamins and minerals) and no infant formula or non-human milk at each study time point and the week immediately prior. The impact of feeding practices on initial hospital length of stay was assessed.

## Sample size

A sample size of 230 women allowed for the percentage of breastfeeding (exclusive or any) to be estimated with a precision of ± 7% or better, with precision defined as a 95% confidence interval (CI) width around the estimated percentage. The sample size allowed for a 10% loss to follow-up and conservatively assumes no gains in precision due to the inclusion of multiple births.

### Ethical considerations

Breastfeeding mothers of late preterm infants were approached in the Neonatal Unit or Postnatal ward and informed consent was obtained for their infant/s. Ethics approval was granted by the Women's and Children's Health Network Human Research Ethics Committee HREC/18/WCHN/064. Site-specific and local governance approvals were obtained at each study site.

### Statistical methods

Descriptive statistics are given as means with standard deviations (SD) or frequencies (percentages) according to the type and distribution of the data. Differences in infant hospital length of stay according to early use of formula was analysed using linear regression, with generalised estimating equations used to account for clustering due to multiple births. An adjustment was made for gestational age, maternal age, multiple birth, site and neonatal intensive care unit admission, all considered potential confounders of the association, with results described as a mean difference with a 95% CI. Associations between maternal and infant characteristics and breastfeeding at 6 weeks, 3 months and 6 months CA were assessed using multivariable logistic regression models (separate models for exclusive breastfeeding and any breastfeeding at each time point), with generalised estimating equations used to account for clustering due to multiple births. Effects are described as odds ratios (OR) with 95% CI. Maternal and infant characteristics were included in multivariable models based on factors associated with increased breastfeeding rates reported in the literature [13]. There was no evidence of collinearity between characteristics or poor model fit in any of the multivariable models considered. Due to low loss to follow-up rates, complete case analyses were used to address missing data. Statistical calculations were performed using R version 4.0.2 (R Foundation for Statistical Computing, Vienna, Austria).

## Results

Two hundred seventy infants born at $34^{+0}$ to $36^{+6}$ weeks gestational age born to 229 mothers were recruited and included in the final analysis (Fig 1).

The mean age of mothers was 31.3 years and most women were born in Australia (76%). A high proportion of women planned to breastfeed their baby prior to delivering (84%) and 64% planned to continue breastfeeding for at least 6 months. Slightly more males were enrolled (58%) and the mean infant birthweight was 2.5Kg. See **Table 1** for complete description of the characteristics of mothers and their infants included in the study.

### Feeding practices

During their initial hospital stay, 97% (n = 261) of infants received breastmilk; 17% (n = 45) received breastmilk only, and 83% (n = 225) received infant formula during their hospital stay; no infants received donor breast milk, **Table 2**. The mean length of stay for infants (n = 264) was 10.9 days (SD 9.4), (Table 1) compared to 8.2 days (SD 7.2) for those receiving breastmilk only (n = 45) and 11.4 (SD 9.7) days for infants that received any formula (n = 216). Adjusting for maternal age, multiple birth, study site and neonatal intensive care admission, the mean difference between the breastmilk only and any formula groups was 5.5 days (95% CI 2.86 to 8.20). The difference was attenuated with further adjustment for gestational age but remained significantly different at 2.9 days (95% CI 0.31 to 5.43, p = 0.03).

On initial discharge home, 74% (n = 200) of the infants were exclusively breastfeeding (Table 2). At 6 weeks CA, 72% (n = 183) of infants were still receiving some breastmilk. By 3 months CA, this was 64% (n = 162), and by 6 months CA, this had fallen further to 53%

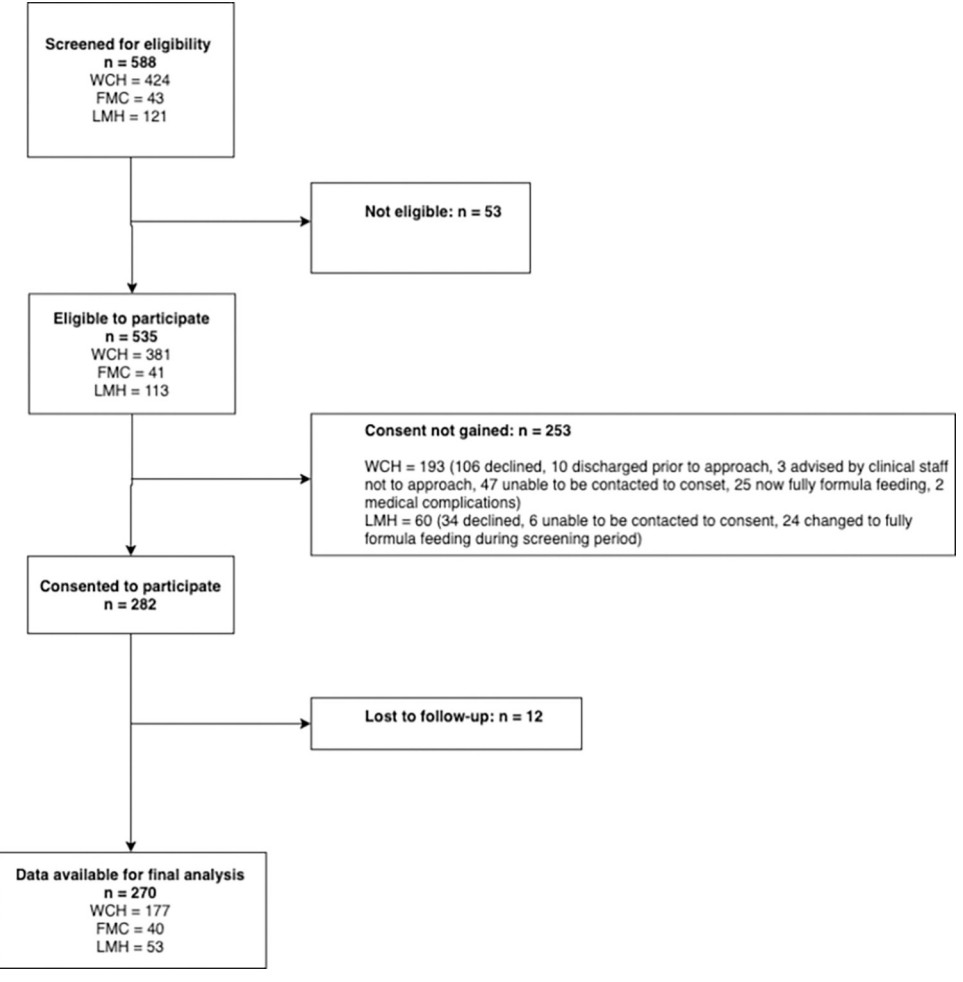

**Fig 1. Participant flow chart–infants.**

(n = 132) infants receiving any breastmilk. Exclusive breastfeeding at 6 weeks CA was 41% (n = 103), and at 3 months CA was at 35% (n = 88), **Table 3**.

### Maternal and infant factors associated with longer duration of breastfeeding

Gestational age was not associated with increased exclusive or partial breastfeeding duration at 6 weeks, 3 months, or 6 months CA, **Table 4**. Maternal intention to breastfeed longer than 6 months was associated with an increased likelihood of receiving exclusive breastmilk at all time points. Receipt of any formula as the first milk feed was associated with a decreased likelihood of exclusive or any breastfeeding at 6 weeks, adjusted OR 0.22 (0.09, 0.53), OR 0.36 (0.14, 0.93), respectively. Care at one study centre was associated with an increased likelihood of any breastfeeding at 6 weeks CA (OR 9.25, 95% CI 1.77 to 48.26); no other centre associations were found at the other time points, Table 4.

## Discussion

In this prospective cohort of South Australian women intending to provide breastmilk to their late preterm infants, exclusive breastfeeding rates at discharge were lower than rates reported

**Table 1. Maternal and infant characteristics***.

| Maternal Characteristics | N = 229 |
|---|:---:|
| Maternal age (years): Mean (SD), (n = 223) | 31 (5) |
| Mother born in Australia, (n = 220) | 167 (76) |
| Mother Aboriginal or Torres Strait Islander, (n = 216) | 9 (4) |
| SEIFA# A+D quintile, (n = 187) | |
| • 1 | 47 (25) |
| • 2 | 26 (14) |
| • 3 | 57 (31) |
| • 4 | 39 (21) |
| • 5 | 18 (10) |
| Multiple birth, (n = 229) | 41 (18) |
| Mode of birth, (n = 224) | |
| • Vaginal delivery | 122 (55) |
| • Emergency caesarean | 71 (32) |
| • Elective caesarean | 31 (14) |
| Length of inpatient stay (days): Median (IQR), (n = 223) | 3 (2–5) |
| Planned duration of breastfeeding, (n = 207) | |
| • ≤6 months | 31 (15) |
| • >6 months | 133 (64) |
| • Undecided | 43 (21) |
| Planned mode of feeding prior to birth, (n = 221) | |
| • Breastfeeding (by breast or bottle) | 186 (84) |
| • Formula Feeding (or mixed feeding) | 30 (14) |
| • Undecided | 5 (2) |
| **Infant Characteristics** | **N = 270** |
| Sex, (n = 270) | |
| Female | 113 (41.9) |
| Male | 157 (58.1) |
| Birthweight (kg): mean (SD), (n = 262) | 2.5 (0.5) |
| Gestational age <36 weeks, (n = 270) | 150 (56) |
| Jaundice (requiring phototherapy), (n = 263) | 116 (44) |
| Hypoglycaemia‡, (n = 262) | 85 (32) |
| Hypothermia within the first 24 hours of age, (n = 262) | 184 (70) |
| Neonatal intensive care unit admission, (n = 262) | 39 (15) |
| Length of inpatient stay (days): Mean (SD), (n = 264) | 10.9 (9.4) |
| Readmission within the first 7 days of initial discharge, (n = 263) | 26 (10) |

*Results are expressed as No. (%) unless otherwise indicated.

#Socio-Economic Indexes for Areas (SEIFA) 1 being the most disadvantaged and 5 the most advantaged. Further information: https://www.abs.gov.au/websitedbs/censushome.nsf/home/seifa

‡Hypoglycaemia <2.0mmol/L <4 hours of age or <2.5mmol/L >4 hours of age—until 24 hours of age.

in term infant populations in Australia [14]. Approximately half of participating women were providing at least some breastmilk at 6 months CA. Further, Formula use as the first milk feed in the hospital and intention to breastfeed >6 months were significant predictors of breastfeeding duration. Further, The use of formula during hospitalisation was associated with a longer length of stay than when only maternal breastmilk was fed to infants after correcting for potential confounders.

**Table 2. Infant feeding practices during initial hospitalisation.**

| Type of feeding and method | N = 270 N (%) |
|---|---|
| Type of first milk feed | |
| • Breastmilk, any | 203 (77.8) |
| • Formula | 58 (22.2) |
| • Missing | 9 (3.3) |
| Types of milk feeds given during the baby's admission | |
| • Breastmilk (mother's own) | 261 (96.7) |
| • Formula (preterm formulation) | 103 (38.1) |
| • Formula (term formulation) | 168 (62.2) |
| • Other (e.g., breastmilk with additives) | 72 (26.7) |
| Types of feeding methods used during the baby's admission | |
| • Breastfeeding (direct) | 251 (93.0) |
| • Bottle | 206 (76.3) |
| • Gavage feeding | 168 (62.2) |
| • Syringe, Finger-feeding | 175 (64.8) |
| • Cup feeding | 5 (1.9) |
| • Supply line | 6 (2.2) |
| • Other (e.g., perfusor feeding) | 54 (20.0) |
| Received intravenous fluid for nutrition or hydration (n = 261) | 113 (43.3) |
| Missing | 9 (3.3) |
| Types of feeds on discharge home | |
| • Breastmilk only (mothers own) | 200 (74.1) |
| • Breastmilk and formula (mixed feeding) | 58 (21.5) |
| • Formula | 41 (15.2) |
| • Other (e.g. breastmilk with additives) | 35 (13.0) |
| Types of feeding methods on discharge home | |
| • Breastfeeding (direct) | 239 (88.5) |
| • Bottle | 175 (64.8) |
| • Gavage feeding | 54 (20.0) |
| • Syringe, Finger-feeding | 21 (7.8) |
| • Supply line | 1 (0.4) |
| • Other | 14 (5.2) |

**Table 3. Breast feeding outcomes at 6 weeks, 3 months and 6 months corrected age.**

| Timepoint | N = 270 N (%) [95% CI] |
|---|---|
| Breastfeeding at 6 weeks, (n = 251): | |
| • Exclusive | 103 (41.0) [34.1, 47.9] |
| • Mixed | 80 (31.9) [25.2, 38.6] |
| • None | 68 (27.1) [20.7, 33.4] |
| Breastfeeding at 3 months, (n = 253): | |
| • Exclusive | 88 (34.8) [28.3, 41.3] |
| • Mixed | 74 (29.2) [22.9, 35.6] |
| • None | 91 (36.0) [29.3, 42.7] |
| Breastfeeding at 6 months, (n = 248): | |
| • Exclusive | 71 (28.6) [22.4, 34.9] |
| • Mixed | 61 (24.6) [18.6, 30.6] |
| • None | 116 (46.8) [39.7, 53.9] |

**Table 4. Multivariable models to determine factors associated with exclusive breastfeeding and any breastfeeding.**

| Variable | 6 weeks | | 3 months | | 6 months | |
| --- | --- | --- | --- | --- | --- | --- |
| | Adjusted OR exclusive breastfeeding (95% CI) | Adjusted OR any breastfeeding (95% CI) | Adjusted OR exclusive breastfeeding (95% CI) | Adjusted OR any breastfeeding (95% CI) | Adjusted OR exclusive breastfeeding (95% CI) | Adjusted OR any breastfeeding (95% CI) |
| *Centre = WCH | 1 (reference) | 1 (reference) | 1 (reference) | 1 (reference) | 1 (reference) | 1 (reference) |
| Centre = FMC | 1.04 (0.33, 3.34) | **9.25 (1.77, 48.26)** | 0.92 (0.29, 2.98) | 2.26 (0.75, 6.83) | 1.20 (0.36, 4.01) | 2.21 (0.55, 8.89) |
| Centre = LMH | 1.24 (0.39, 3.90) | 0.48 (0.14, 1.64) | 1.49 (0.55, 4.09) | 0.54 (0.17, 1.74) | 0.85 (0.26, 2.83) | 0.66 (0.21, 2.08) |
| ‡Plan to breastfeed >6mths | **4.98 (2.39, 10.40)** | **10.79 (4.37, 26.63)** | **4.67 (2.20, 9.92)** | **9.21 (4.26, 19.92)** | **6.64 (2.67, 16.47)** | **6.92 (3.26, 14.66)** |
| Child sex = male | 0.72 (0.36, 1.42) | **0.31 (0.12, 0.79)** | 0.78 (0.40, 1.54) | 0.65 (0.30, 1.44) | 0.65 (0.31, 1.35) | 0.69 (0.34, 1.40) |
| NICU = yes | 0.89 (0.40, 2.00) | 1.32 (0.41, 4.26) | 0.84 (0.36, 1.95) | 0.83 (0.32, 2.12) | 0.89 (0.35, 2.23) | 0.84 (0.40, 1.77) |
| Formula as first milk feed | **0.22 (0.09, 0.53)** | **0.36 (0.14, 0.93)** | 0.51 (0.22, 1.17) | 0.51 (0.20, 1.30) | 0.43 (0.16, 1.15) | 0.67 (0.26, 1.74) |
| feed = formula only | | | | | | |
| #SEIFA 1 | 1 (reference) | 1 (reference) | 1 (reference) | 1 (reference) | 1 (reference) | 1 (reference) |
| SEIFA 2 | 0.98 (0.34, 2.83) | 1.00 (0.28, 3.61) | 1.03 (0.37, 2.91) | 2.03 (0.59, 7.03) | 0.80 (0.29, 2.26) | 1.33 (0.42, 4.25) |
| SEIFA 3 | 1.47 (0.62, 3.53) | 2.42 (0.79, 7.46) | 1.53 (0.62, 3.77) | 2.34 (0.86, 6.35) | 0.93 (0.37, 2.34) | 1.64 (0.69, 3.89) |
| SEIFA 4 | 1.64 (0.58, 4.61) | 1.80 (0.48, 6.80) | 1.88 (0.69, 5.11) | 2.09 (0.67, 6.54) | 1.05 (0.35, 3.18) | 1.44 (0.51, 4.09) |
| SEIFA 5 | 1.04 (0.25, 4.33) | 5.13 (0.92, 28.67) | 2.02 (0.50, 8.12) | **11.19 (2.40, 52.05)** | 0.99 (0.23, 4.26) | 4.33 (0.73, 25.52) |

Results reaching statistical significance are in bold

**Abbreviations:** CI, Confidence interval; OR, odds ratio; WCH, Women's and Children's Hospital; FMC, Flinders Medical Centre; LMH, Lyell McEwin Hospital; NICU, neonatal intensive care unit; SEIFA, Socio-Economic Indexes for Areas

*Global p-value for Centre: (6 weeks) OR exclusive P = 0.93 and OR mixed or exclusive P = 0.009; (3 months) OR exclusive P = 0.71 and OR mixed or exclusive P = 0.14; (6 months) OR exclusive P = 0.91 and OR mixed or exclusive P = 0.13

‡'Undecided' and ≤6 months combined as reference group due to small numbers

#Socio-Economic Indexes for Areas (SEIFA) 1 being the most disadvantaged and 5 the most advantaged. Global p-value for SEIFA: (6 weeks) OR exclusive P = 0.81 and OR mixed or exclusive P = 0.27; (3 months) OR exclusive P = 0.64 and OR mixed or exclusive P = 0.04; (6 months) OR exclusive P = 0.99 and OR mixed or exclusive P = 0.51

Breastmilk is the optimal nutrition for infants and critical for late preterm infants to support overall growth and development, gut maturation and immune protection [15, 16]. In our cohort study of women who intended to provide at least some breastmilk to their babies, 80% of their babies received any formula as supplementary nutrition. This is consistent with the limited existing literature reporting early nutrition practices in late preterm infants, which suggests that formula use is common during the neonatal admission [15].

Our findings may reflect the frequent delays in establishing breastfeeding in this population, potentially resulting in the use of formula to avoid hypoglycaemia, dehydration and to support the high nutrient requirements for postnatal growth. Interestingly, formula use compared to the use of maternal breastmilk only was associated with a longer length of hospital stay. This association remained after accounting for factors indicative of the complexity of care the infants required (e.g., gestational age, neonatal intensive care admission, and multiple births). Further exploration of this association is warranted, as it may reflect factors not accounted for in our models, such as maternal illness. Nevertheless, formula use during the neonatal admission may be modifiable with interventions such as pasteurised donor human milk or oral dextrose preparations [17]. There remains little high-quality evidence to guide the clinical management of nutrition support in late preterm infants. Further research is warranted to understand the impact of adopting alternative strategies to maximise nutritional outcomes in this population.

Previous studies have reported that approximately 60% of late preterm infants leave the hospital being breastfed exclusively, and the overall duration of breastfeeding is shorter when compared with infants born at term [9, 18, 19]. Our study found higher rates of exclusive breastfeeding at discharge (74%), however, this dropped substantially to 35% by 3 months CA. Maternal intention to breastfeed for >6 months was the most consistent factor associated with a longer duration of breastfeeding across all time periods. This is consistent with previous research [20], predominantly in term infants, and points to the need to promote breastfeeding antenatally to help women clarify their intentions. Our findings suggest there is room to improve support for women after they leave the hospital. Previous research has identified a lack of support from health care professionals in hospital and after discharge home and perceptions about low breastmilk supply as key barriers to breastfeeding late preterm infants [21]. These barriers require addressing and are identified as a priority by parents and care providers [22, 23].

Strengths of our study include recruitment from multiple hospital sites with diverse populations, prospective collection of data minimising recall bias and minimal attrition to 6 months CA. Our study was designed to examine factors associated with breastfeeding duration in women who intended to provide at least some breastmilk to their infant, so there may be limitations with the generalisability of our findings to the whole late preterm infant population. Feeding outcomes post infant discharge were collected by maternal report, as were other post-discharge outcomes; thus, the findings could be subject to reporting bias. Further, residual confounding remains a risk in the analyses examining the length of stay and factors associated with breastfeeding duration, as not all relevant covariates could be included in the model.

Our findings demonstrate that exclusive breastfeeding rates were low in a cohort of women intending to breastfeed their late preterm infants, with less than half providing breastmilk at 6 months corrected age. early feeding practices including the use of formula as the first feed in hospital, and intention to breastfeed >6 months were significant predictors of breastfeeding duration in this population. These findings highlight the difficulties women experiencing late preterm birth face in establishing and sustaining breastfeeding and the need for improved support during initial hospitalisation and in the first few months following birth.

## Author Contributions

**Conceptualization:** Amy Keir, Carmel T. Collins, Andrew J. McPhee, Jojy Varghese, Scott Morris, Karen P. Best.

**Data curation:** Thomas R. Sullivan, Shalem Leemaqz.

**Formal analysis:** Alice Rumbold, Thomas R. Sullivan, Shalem Leemaqz.

**Methodology:** Amy Keir, Carmel T. Collins, Thomas R. Sullivan, Maria Makrides, Karen P. Best.

**Project administration:** Amy Keir, Carmel T. Collins.

**Resources:** Jojy Varghese, Scott Morris, Thomas R. Sullivan, Maria Makrides, Karen P. Best.

**Supervision:** Alice Rumbold, Carmel T. Collins, Philippa Middleton, Maria Makrides, Karen P. Best.

**Validation:** Alice Rumbold, Thomas R. Sullivan.

**Writing – original draft:** Amy Keir.

**Writing – review & editing:** Amy Keir, Alice Rumbold, Carmel T. Collins, Andrew J. McPhee, Jojy Varghese, Scott Morris, Thomas R. Sullivan, Shalem Leemaqz, Philippa Middleton, Maria Makrides, Karen P. Best.

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
