## [Decision Letter · Decision Letter 0]

7 Feb 2022

PONE-D-21-31091Breastfeeding outcomes in late preterm infants: a multi-centre prospective cohort studyPLOS ONE

Dear Dr.Amy Keir/>Thank you for submitting your manuscript to PLOS ONE. After careful consideration, we feel that it has merit but does not fully meet PLOS ONE’s publication criteria as it currently stands. Therefore, we invite you to submit a revised version of the manuscript that addresses the points raised during the review process.

ACADEMIC EDITOR: The authors come up with a study problem of significance to the existing literature. Moreover, it has been stated using understandable English. However, there are minor typos and confusing statements as the reviewers raised. Most importantly, kindly address all the reviewers concerns as they would further enrich your manuscript. Please submit your revised manuscript by Mar 24 2022 11:59PM. If you will need more time than this to complete your revisions, please reply to this message or contact the journal office at plosone@plos.org. Please include the following items when submitting your revised manuscript:A rebuttal letter that responds to each point raised by the academic editor and reviewer(s). You should upload this letter as a separate file labeled 'Response to Reviewers'.A marked-up copy of your manuscript that highlights changes made to the original version. You should upload this as a separate file labeled 'Revised Manuscript with Track Changes'.An unmarked version of your revised paper without tracked changes. You should upload this as a separate file labeled 'Manuscript'.If applicable, we recommend that you deposit your laboratory protocols in protocols.io to enhance the reproducibility of your results. Protocols.io assigns your protocol its own identifier (DOI) so that it can be cited independently in the future. For instructions see: https://journals.plos.org/plosone/s/submission-guidelines#loc-laboratory-protocols. Additionally, PLOS ONE offers an option for publishing peer-reviewed Lab Protocol articles, which describe protocols hosted on protocols.io. Read more information on sharing protocols at https://plos.org/protocols?utm_medium=editorial-email&utm_source=authorletters&utm_campaign=protocols.

We look forward to receiving your revised manuscript.

Kind regards,

Wubet Alebachew Bayih, M.Sc.

Academic Editor

PLOS ONE

Journal Requirements:

Reviewers' comments:

Reviewer's Responses to Questions

**Comments to the Author**

1. Is the manuscript technically sound, and do the data support the conclusions?

Reviewer #1: Yes

Reviewer #2: Yes

2. Has the statistical analysis been performed appropriately and rigorously? 

Reviewer #1: Yes

Reviewer #2: Yes

3. Have the authors made all data underlying the findings in their manuscript fully available?

Reviewer #1: Yes

Reviewer #2: Yes

4. Is the manuscript presented in an intelligible fashion and written in standard English?

Reviewer #1: Yes

Reviewer #2: Yes

5. Review Comments to the Author

Reviewer #1: Find my comments here below

This manuscript is interesting to a global audience having an Australian perspective. However, I see the need for some clarifications, and hopefully my comments will help you to improve your manuscript.

Methods

1.Page 5, line 81 & 82: Indicate the specific recruitment date, completed date, and data collection finalised date

2.Page 6, line 107: Is that any human milk or their mothers’ milk? Does this include donor breast milk?

3.State the way you used to take consents for under 18 years old participants under ethical consideration section

4. Was the questionnaire validated in the local context and how?

5.Have you tested multicollinearity and the model fitness? If not reason? If yes, indicate it clearly.

Results

6.Table 1: what do you mean “born in Australia, (n=220)”? since your study was conducted completely @Australia.

Reviewer #2: Thank you Dr.Wubet for inviting me to review this manuscript.

This article describes the breastfeeding outcome of late preterm infants. The stated goal of the study was to explore factors associated with breastfeeding duration and describe infant feeding practices during initial hospitalization and up to 6 months corrected age in infants born late preterm with mothers intending to breastfeed.

The article is generally well-written and interesting. If possible, the methods and analysis should be reviewed by a university biostatistician.

It is important to spread awareness of exclusive breastfeeding to create opportunities in improving infants' health. Therefore, this study is very important and needs to be published.

Suggestions, questions

Abstract

Avoid abbreviations in the abstract

Background -It would be better if you add the burden of the problem and the rationales why you are doing this research

In the methods and materials section of the abstract- The type of model used to determine the factors associated with breastfeeding duration was not mentioned. I didn't understand why you used method and material? What materials did you use in this study?

The conclusion section of the abstract – I have confused by the findings stated in your result section and conclusion. Receipt of formula milk was associated with BF (breastfeeding) duration in the result section but in the conclusion part, early feeding practice and intention to breastfeed were predictors of BF duration.?

Introduction, method, and material -I am satisfied with this section of the manuscript.

Outcome: line 104 - initial hospital length of stay and exclusive or any breastfeeding at 6

weeks, 3 months, and 6 months CA. what parameters do you use to categorize the outcome variable into 6 weeks,3 months, and six months?

- I couldn't find the adjusted odds ratio value in your multivariate logistic regression model to identify the factors associated with breastfeeding duration. if I was not mistaken, how could you conclude the factors associated with breastfeeding duration without making an adjusted odds ratio? you have made only bivariate logistic regression based on table 4.

Result –you have made too long and interrupted tables, and lacks description

Table 3- breastfeeding at 6 months, (n=??? 248) - I am not clear about it? Generally, tables are not prepared in the way of attracting readers.

Discussion

Line 212- Line 215 this paragraph lacks reference.

Line 218 – L 220. This statement lacks justification about exclusive breastfeeding at discharge related to an increase from 60 % to 74% and decreasing to 33% by 3 months of CA.

Maternal intention to breastfeed>6 months is the most consistent predictor of breastfeeding duration. This factor was not well discussed with other research findings.

6. PLOS authors have the option to publish the peer review history of their article (what does this mean?). If published, this will include your full peer review and any attached files.

Reviewer #1: **Yes: **Tamirat Getachew

Reviewer #2: No

---

## [Author Response · Author response to Decision Letter 0]

4 Mar 2022

Dr Wubet Alebachew Bayih

PLOS One Editorial Office

RE: PONE-D-21-31091: Breastfeeding outcomes in late preterm infants: a multi-centre prospective cohort study

Dear Dr Wubet

Thank you for considering our Original Research paper listed above for publication in PLOS One.

Please find our rebuttal letter that responds to each point raised by the academic editor and reviewer(s).

Reviewer #1: 

1.Page 5, line 81 & 82: Indicate the specific recruitment date, completed date, and data collection finalised date

All dates updated to exact date.

2.Page 6, line 107: Is that any human milk or their mothers’ milk? Does this include donor breast milk?

Methods clarified and wording updated;

Exclusive breastfeeding was defined as a baby receiving only breastmilk and medications, including oral rehydration solutions, vitamins and minerals, but no infant formula or non-human milk at each study time point and the week immediately preceding them.

3.State the way you used to take consents for under 18 years old participants under ethical consideration section

Ethical Considerations section updated;

Mothers of late preterm were approached in the Neonatal Units or Postnatal ward and informed consent was obtained for their infant/s. Ethics approval was granted by the Women’s and Children’s Health Network Human Research Ethics Committee HREC/18/WCHN/064. Site-specific and local governance approvals were obtained at each study site.

4. Was the questionnaire validated in the local context and how?

There wasn’t really one questionnaire for the study. Data at enrolment was obtained via maternal report in person, telephone or surveys that were designed as electronic data capture instruments. Short follow up surveys were sent to women at study timepoints (6 weeks, 3 months and 6 months) so that women could report on their breastfeeding practices remotely via text or email. Methods updated for clarity.

5.Have you tested multicollinearity and the model fitness? If not reason? If yes, indicate it clearly.

Relationships between measures of breastfeeding over time, or between exclusive breastfeeding and any breastfeeding at each time point were considered in separate statistical models. We have now explained in the methods that “There was no evidence of collinearity between characteristics or poor model fit in any of the multivariable models considered.” We have also clarified in the statistical methods that measures of breastfeeding (exclusive or any) were analysed in separate models at each time point (hence there were no concerns around collinearity in the breastfeeding outcomes over time).

6. Table 1: what do you mean “born in Australia, (n=220)”? since your study was conducted completely @Australia.

This is listed under maternal characteristics and refers to mothers born in Australia, table line headings have been updated for clarity.

Reviewer #2: 

Abstract - Avoid abbreviations in the abstract

Abbreviations removed unless used multiple times, i.e. corrected age.

Background -It would be better if you add the burden of the problem and the rationales why you are doing this research

Further explanation added line 57 and 58: ‘Breastmilk is the optimal nutrition to support overall growth and development, gut maturation and immune protection. Complications relating to poor feeding are major contributors to the health burden in this population and one of the leading causes of extended hospital stay and readmission.’

In the methods and materials section of the abstract- The type of model used to determine the factors associated with breastfeeding duration was not mentioned. I didn't understand why you used method and material? What materials did you use in this study?

In the methods it states that “Associations between maternal and infant characteristics and breastfeeding at 6 weeks, 3 months and 6 months CA were assessed using multivariable logistic regression models (separate models for exclusive breastfeeding and any breastfeeding at each time point), with generalised estimating equations used to account for clustering due to multiple births. Effects are described as odds ratios (OR) with 95% CI. Maternal and infant characteristics were included in multivariable models based on factors associated with increased breastfeeding rates reported in the literature.”

Heading changed to Methods.

The conclusion section of the abstract – I have confused by the findings stated in your result section and conclusion. Receipt of formula milk was associated with BF (breastfeeding) duration in the result section but in the conclusion part, early feeding practice and intention to breastfeed were predictors of BF duration?

Sentence revised for clarity: Early feeding practices including the use of formula in hospitals, and intention to breastfeed >6 months were significant predictors of breastfeeding duration in this population.

Outcome: line 104 - initial hospital length of stay and exclusive or any breastfeeding at 6 weeks, 3 months, and 6 months CA. what parameters do you use to categorize the outcome variable into 6 weeks,3 months, and six months?

Key outcomes were exclusive or any breastfeeding at 6 weeks, 3 months, and 6 months. Surveys were sent to women at these timepoints. Initial length of stay was assessed but not in relation to feeding practices at 6 weeks, 3 months and 6 months. observation. This paragraph has been edited for clarity.

- I couldn't find the adjusted odds ratio value in your multivariate logistic regression model to identify the factors associated with breastfeeding duration. if I was not mistaken, how could you conclude the factors associated with breastfeeding duration without making an adjusted odds ratio? you have made only bivariate logistic regression based on table 4.

All odds ratios in Table 4 are from multivariable logistic regression models. For example, the odds ratios provided for centre are adjusted for duration planned to breastfeed, sex, NICU admission, formula as first milk feed and SEIFA. To make this clearer, we have now labelled the effect estimate as “Adjusted OR” in the table.

Result –you have made too long and interrupted tables, and lacks description

Tables re-formatted and description added.

Table 3- breastfeeding at 6 months, (n=??? 248) - I am not clear about it? Generally, tables are not prepared in the way of attracting readers.

Table 3 corrected and all tables modified for readability.

Discussion

Line 212- Line 215 this paragraph lacks reference.

This paragraph has been modified: Nevertheless, formula use during the neonatal admission may be modifiable with interventions such as pasteurised donor human milk or oral dextrose preparations. There remains little high-quality evidence to guide the clinical management of nutrition support in late preterm infants. Further research is warranted to understand the impact of adopting alternative strategies to maximise nutritional outcomes in this population.

Line 218 – L 220. This statement lacks justification about exclusive breastfeeding at discharge related to an increase from 60 % to 74% and decreasing to 33% by 3 months of CA.

This paragraph has been modified.

Maternal intention to breastfeed>6 months is the most consistent predictor of breastfeeding duration. This factor was not well discussed with other research findings.

Line 226 to line 234 in revised manuscript discusses intention to breastfeed: Maternal intention to breastfeed for >6 months was the most consistent factor associated with a longer duration of breastfeeding across all time periods. This is consistent with previous research, predominantly in term infants, and points to the need to promote breastfeeding antenatally to help women clarify their intentions. Our findings suggest there is room to improve support for women after they leave the hospital. Previous research has identified a lack of support from health care professionals in hospital and after discharge home and perceptions about low breastmilk supply as key barriers to breastfeeding late preterm infants. These barriers require addressing and are identified as a priority by parents and care providers. 

A clean and marked up copy of our revised manuscript has been uploaded to Editorial Manager.

We look forward to hearing back from you.

Yours sincerely

Associate Professor Amy Keir

On behalf of the Authorship Group

---

## [Decision Letter · Decision Letter 1]

6 Apr 2022

PONE-D-21-31091R1Breastfeeding outcomes in late preterm infants: a multi-centre prospective cohort studyPLOS ONE

Dear Dr. Amy Keir,

Thank you for submitting your manuscript to PLOS ONE. After careful consideration, we feel that it has merit but does not fully meet PLOS ONE’s publication criteria as it currently stands. Therefore, we invite you to submit a revised version of the manuscript that addresses the points raised during the review process.

ACADEMIC EDITOR: The authors have made significant improvement to the raised concerns. However, there are still some issues that need to be considered as of reviewer 2. 

We look forward to receiving your revised manuscript.

Kind regards,

Wubet Alebachew Bayih, M.Sc.

Academic Editor

PLOS ONE

Journal Requirements:

Reviewers' comments:

Reviewer's Responses to Questions

**Comments to the Author**

1. If the authors have adequately addressed your comments raised in a previous round of review and you feel that this manuscript is now acceptable for publication, you may indicate that here to bypass the “Comments to the Author” section, enter your conflict of interest statement in the “Confidential to Editor” section, and submit your "Accept" recommendation.

Reviewer #1: All comments have been addressed

Reviewer #2: (No Response)

2. Is the manuscript technically sound, and do the data support the conclusions?

Reviewer #1: Yes

Reviewer #2: Yes

3. Has the statistical analysis been performed appropriately and rigorously? 

Reviewer #1: Yes

Reviewer #2: Yes

4. Have the authors made all data underlying the findings in their manuscript fully available?

Reviewer #1: Yes

Reviewer #2: Yes

5. Is the manuscript presented in an intelligible fashion and written in standard English?

Reviewer #1: Yes

Reviewer #2: Yes

6. Review Comments to the Author

Reviewer #1: (No Response)

Reviewer #2: The authors were not fully addressed my concerns especially in the abstract part of the manuscript. For instance,

1. In the methods and materials section of the abstract- The type of model used to determine the factors associated with breastfeeding duration was not mentioned. Subheading in the abstract is still method and material.

2. The conclusion section of the abstract – I have confused by the findings stated in your result section and conclusion. Receipt of formula milk was associated with BF (breastfeeding) duration in the result section but in the conclusion part, early feeding practice and intention to breastfeed were predictors of BF duration?

3. Table 1 and table 3 needs description in the frequency column

7. PLOS authors have the option to publish the peer review history of their article (what does this mean?). If published, this will include your full peer review and any attached files.

Reviewer #1: **Yes: **Tamirat Getachew

Reviewer #2: No

---

## [Author Response · Author response to Decision Letter 1]

8 Apr 2022

Thank you for your email received 7th April 2022 with additional comments from Reviewer 2. We appreciate the thorough review and feedback to improve our manuscript. Please find our responses below:

Reviewer #2: The authors were not fully addressed my concerns especially in the abstract part of the manuscript. For instance,

1. In the methods and materials section of the abstract- The type of model used to determine the factors associated with breastfeeding duration was not mentioned. Subheading in the abstract is still method and material.

A description of the model has been added to abstract methods: “Associations between maternal and infant characteristics and breastfeeding at 6 weeks, 3 months and 6 months CA were assessed using multivariable logistic regression models.”

Subheading updated to “methods”.

2. The conclusion section of the abstract – I have confused by the findings stated in your result section and conclusion. Receipt of formula milk was associated with BF (breastfeeding) duration in the result section but in the conclusion part, early feeding practice and intention to breastfeed were predictors of BF duration?

Abstract conclusion updated for clarity to: “Formula as the first milk feed and intention to breastfeed >6 months were significant predictors of breastfeeding duration.

3. Table 1 and table 3 needs description in the frequency column 

Description added to frequency column in Table 1 and 3.

---

## [Decision Letter · Decision Letter 2]

22 Jul 2022

Breastfeeding outcomes in late preterm infants: a multi-centre prospective cohort study

PONE-D-21-31091R2

Dear Dr. Keir,

We’re pleased to inform you that your manuscript has been judged scientifically suitable for publication and will be formally accepted for publication once it meets all outstanding technical requirements.

Kind regards,

Ralph C. A. Rippe, Ph.D.

Academic Editor

PLOS ONE

Reviewers' comments:

Reviewer's Responses to Questions

**Comments to the Author**

1. If the authors have adequately addressed your comments raised in a previous round of review and you feel that this manuscript is now acceptable for publication, you may indicate that here to bypass the “Comments to the Author” section, enter your conflict of interest statement in the “Confidential to Editor” section, and submit your "Accept" recommendation.

Reviewer #1: All comments have been addressed

Reviewer #2: All comments have been addressed

2. Is the manuscript technically sound, and do the data support the conclusions?

Reviewer #1: Yes

Reviewer #2: Yes

3. Has the statistical analysis been performed appropriately and rigorously? 

Reviewer #1: Yes

Reviewer #2: Yes

4. Have the authors made all data underlying the findings in their manuscript fully available?

Reviewer #1: (No Response)

Reviewer #2: Yes

5. Is the manuscript presented in an intelligible fashion and written in standard English?

Reviewer #1: Yes

Reviewer #2: Yes

6. Review Comments to the Author

Reviewer #1: (No Response)

Reviewer #2: Thank you for addressing all my concerns . Now, the manuscript is suitable for publication to plos one

7. PLOS authors have the option to publish the peer review history of their article (what does this mean?). If published, this will include your full peer review and any attached files.

Reviewer #1: **Yes: **Tamirat Getachew

Reviewer #2: **Yes: **Natnael Moges

---

## [Editor Report · Acceptance letter]

5 Aug 2022

PONE-D-21-31091R2 

Breastfeeding outcomes in late preterm infants: a multi-centre prospective cohort study 

Dear Dr. Keir:

I'm pleased to inform you that your manuscript has been deemed suitable for publication in PLOS ONE. Congratulations! Your manuscript is now with our production department. 

Kind regards, 

on behalf of

Dr. Ralph C. A. Rippe 

Academic Editor

PLOS ONE